# Extrapolating Multilingual Language Understanding Models as Multilingual Language Generators

**Bohong Wu[1,2], Fei Yuan[3*], Hai Zhao[1,2†], Lei Li[4], Jingjing Xu[3‡]**

[1] Department of Computer Science and Engineering, Shanghai Jiao Tong University
[2] Key Laboratory of Shanghai Education Commission for Intelligent Interaction
and Cognitive Engineering, Shanghai Jiao Tong University, Shanghai, China
[3] Shanghai Artificial Intelligence Laboratory [4] Carnegie Mellon University
chengzhipanpan@sjtu.edu.cn, yuanfei@pjlab.org.cn,
zhaohai@cs.sjtu.edu.cn, leili@cs.cmu.edu, jingjingxupku.02@gmail.com

## Abstract

Multilingual understanding models (or encoder-based), pre-trained via masked language modeling, have achieved promising results on many language understanding tasks (e.g., mBERT). However, these models are not capable of generating high-quality text compared with decoder-based causal language models. Can we transform a pre-trained language understanding model into an effective language generation model? We propose a **S**emantic-**G**uided **A**lignment-then-Denoising (SGA) approach to adapt a multilingual encoder to a multilingual generator with a small number of additional parameters. Experiments show that the proposed approach is an effective adaption method, outperforming widely-used initialization-based methods with gains of 9.4 BLEU on machine translation, 8.1 Rouge-L on question generation, and 5.5 METEOR on story generation on XLM-R$_{large}$. On the other hand, we observe that XLM-R is still inferior to mBART in supervised settings despite better results on zero-shot settings, indicating that more exploration is required to make understanding models strong generators. Our code is available at https://github.com/chengzhipanpan/XLMR4MT.

## 1 Introduction

Multilingual encoder-based models (e.g., mBERT (Pires et al., 2019), XLM-R (Conneau et al., 2020)), pre-trained via masked language modeling, have demonstrated strong performance on a wide range of understanding tasks (Conneau et al., 2018; Liang et al., 2020; Hu et al., 2020). Existing multilingual pre-trained models can be classified into two settings: autoregressive models (Liu et al., 2020; Xue et al., 2021; Scao et al., 2022) and non-autoregressive models (Pires et al., 2019; Conneau

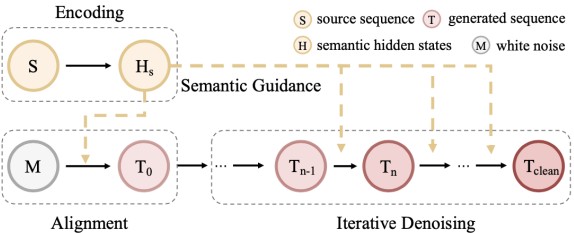

Figure 1: An overview of semantic-guided generation using pre-trained understanding models. The encoding step is responsible for mapping the source input into a shared space that supervises the following generation. By taking the source input and a blank sentence (white noise) as input, the alignment stage generates target tokens simultaneously. Then, we feed the source representations and the generated sequence into the denoising stage for NAR denoising. The denoising step is performed iteratively until the generated text keeps unchanged or reaches the maximum loop.

et al., 2020; Ouyang et al., 2021). Typically, the AR framework, where a target sequence is generated from left to right, succeeds in multilingual generation tasks (Chen et al., 2022; Qi et al., 2018). As a comparison, encoder-based models are NAR models that are usually limited to understanding tasks (Conneau et al., 2018). Despite superior understanding results over AR models, these NAR models still struggle to handle a wide range of multilingual generation tasks. However, NAR models still have obvious advantages in generation efficiency and decoding flexibility (Gu et al., 2018; Qian et al., 2021; Huang et al., 2021; Ghazvininejad et al., 2019; Saharia et al., 2020), which enables generating multiple tokens at one time in arbitrary order. Considering these strengths, this paper aims to explore methods to make multilingual encoder-based models better generators with a small number of new parameters.

There is limited research focusing on empowering understanding models with the generation ability. Traditional methods usually use pre-trained

---
[*]This work was partially supported by the National Key R&D Program of China (NO.2022ZD0160100)

[†]This work was also partially supported by Joint Research Project of Yangtze River Delta Science and Technology Innovation Community (No. 2022CSJGG1400).

[‡]Corresponding author.

encoders as initializers for AR models in various monolingual generation tasks (Su et al., 2021). Despite promising results, it does not satisfy our target that fixes pre-trained parameters to build a unified model for any language tasks. More recently, researchers have focused on learning-free approaches (Wang and Cho, 2019; Kumar et al., 2022b; Qin et al., 2022). One typical approach is iteratively choosing tokens to mask and sampling proposals using energy models (Mireshghallah et al., 2022), resulting in surprising high latency. Furthermore, these learning-free methods are usually limited to controllable generation and are still inferior in handling complicated tasks like machine translation. Unlike these monolingual studies, adapting multilingual understanding models to multilingual generation has its own challenges: semantic constraints under conditional generation where the generation process should follow the semantic constraints given source texts in any language, and parameter efficiency constraints where a single model can serve text generation in any language.

We propose a semantic-guided approach to address these challenges, with a two-stage generation process: alignment-then-denoising. The two stages share the same pre-trained parameters and only add a small number of new prompt parameters for adaptation. Given that masked language modeling (MLM) is also a denoising objective, existing multilingual pre-trained models can be naturally adapted to good denoisers. Therefore, we introduce a denoising stage into our framework. The whole generation process is shown in Figure 1. The encoding part maps the source input into a shared space that supervises the following generation. By taking the source input and a blank sentence as input, the alignment stage generates target tokens simultaneously. We feed the source representations and the generated sequence into the denoising module for NAR denoising. The denoising step is performed iteratively until the generated text keeps unchanged or the maximum loop is reached.

Experiments demonstrate that our model has achieved better results on various generation tasks than traditional fine-tuning-based approaches that directly use NAR pre-trained models as initialization, with gains of 9.4 BLEU on machine translation, 8.1 Rouge-L on question generation, and 5.5 METEOR on story generation on XLM-R$_{large}$. More promisingly, our method has achieved im-

pressive zero-shot cross-lingual ability in translation tasks, outperforming a multilingual AR adaptation model, mGPT + MSP by a large margin. On the other hand, we also notice the gap between XLM-R and AR models. Generally, XLM-R with fine-tuning is largely inferior to mBART fine-tuning. With our methods, the gap is largely reduced but still exists. In future work, we would like to explore pre-training methods to make multilingual understanding models better generators.

Our contributions can be summarized as follows:

- We propose an efficient adaptation method to make multilingual understanding models better generators in a parameter-efficient way.

- We present a semantic-guided denoiser, which can efficiently improve the generation quality.

- Experiments show that our proposed method outperforms traditional initialization-based adaptation methods by a large margin.

## 2 Related Work

In this section, we review the related studies including parameter-efficient adaptation, adapting encoder models as generators, and non-autoregressive generation.

**Parameter efficient adaptation** Pre-trained language models (PLMs) (Devlin et al., 2019; Liu et al., 2019; Clark et al., 2020; Conneau et al., 2020) have achieved overwhelming performance in a variety of downstream tasks. Parameter-efficient tuning is a hot research direction to adapt PLMs to downstream tasks with only training on a few parameters. Adapter-based methods (Bapna and Firat, 2019) are one of the popular parameter-efficient approaches. Recent studies (Üstün et al., 2021; Cooper Stickland et al., 2021) proposed to use adapters on the top of an mBART (Liu et al., 2020) model, enabling a flexible and well-performed method for plug-and-play translation. More recently, prefix tuning and other prompt-based methods (Li and Liang, 2021; Lester et al., 2021; Liu et al., 2022; Tan et al., 2022) have proved to be extremely helpful, and can easily support mixed-task inference as it does not require changing the architecture of PLMs. In this work, we follow this research thread for efficient adaptation and apply prompt-based approaches in our work.

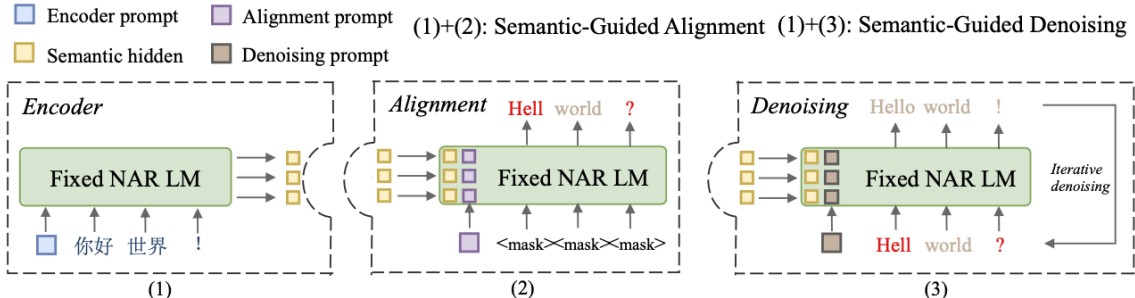

Figure 2: An overview of the generation units. All units share the same pre-trained parameters and individual prompt parameters (shown in blue square, purple square, and brown square). The encoder maps the source input into a sequence of hidden representations (yellow), which are then fed into a decoder and a denoiser for target generation. The alignment unit is responsible for generating a piece of target text. The denoiser is responsible for refining the generated text.

**Adapting encoder-based models for generation** Previous works proposed to use pre-trained understanding models to initialize encoder-decoder models (Chen et al., 2021; Ma et al., 2021) or use the contextualized embeddings produced by understanding models as more aligned inputs for generation (Xu et al., 2021a). With the trend of scaling up models in recent years, it gradually becomes impossible to fine-tune the whole language model in each language direction. In addition, there have been several learning-free methods that adopt encoder models as energy scorers for controllable text generation (Mireshghallah et al., 2022; Kumar et al., 2022a). Although these methods do not need to fine-tune the pre-trained model, they require multi-steps of sampling and refinement, resulting in surprising inference latency.

**Non-autoregressive generation** Our work aims at adapting multilingual encoders to multilingual generators, instead of developing a NAR architecture like previous NAR literature does. Therefore, there is a lot of difference between our work with previous NAR studies. Despite different motivations, our implementation also uses several NAR techniques, like CTC (Graves et al., 2006; Libovický and Helcl, 2018) and Mask-Predict (Ghazvininejad et al., 2019). For clarification, we also review the thread of NAR generation. Single-step NAR generation is a popular research direction that generates text at one time. To mitigate the gap between single-step NAR methods and AR methods, researchers have proposed alignment-based methods (Libovický and Helcl, 2018; Ghazvininejad et al., 2020; Du et al., 2021) or glancing-based methods (Qian et al., 2021). As a compromise, iterative NAR methods can provide

both comparable performance and better latency with AR baselines (Lee et al., 2018; Ghazvininejad et al., 2019; Huang et al., 2021; Saharia et al., 2020). For example, SUNDAE (Savinov et al., 2021) proposed step-unrolled denoising, and achieved good performance in both machine translation and text infilling. Similar iterative idea has been adopted at recent diffusion models (Li et al., 2022; Gong et al., 2022). In this work, we adopt an iterative decoding idea to take advantage of the denoising abilities of encoder-based models which are pre-trained with denoising objectives.

## 3 Notation and Background

**Prompt tuning** For efficient adaption, we follow mGPT+MSP (Tan et al., 2022) and use prompt tuning to adapt an existing pre-trained NAR pre-trained multilingual understanding models to generators. Formally, we denote $K^l$ and $V^l$ as the key-value pairs in the $l$-th Transformer layer. The introduced prompt-tuning parameters are $(K, V)$ pairs, which will be concatenated with current key-value pairs during training and inference. In prompt tuning, we denote the forward pass of a pre-trained LM as $f_{LM}(\theta_p, X)$, which accepts two inputs including prompt parameters $\theta_p$ and the source sequence $X$.

## 4 SGA Approach

### 4.1 Overview

Based on a fixed multilingual understanding model, Figure 2 presents the overview of our proposed SGA, which contains three stages, including semantic encoding, alignment and then denoising. Section 4.2 presents the semantic encoding unit, which

maps sentences in all languages to a unified space. Section 4.3 presents the alignment unit, which generates a target sentence. Section 4.3 presents the denoising unit, which refines the generated sentences under the guidance of semantics.

## 4.2 Semantic Encoding

Suppose a pre-trained multilingual language model with $L$ layers. we define prompt parameters $\theta_{p_s} = (K_s^{1:L}, V_s^{1:L})$ for all layers. These parameters are then concatenated with the key-value pairs of attention in each layer to extract the hidden representation of the source sequence $X = [x_1, x_2, ..., x_t]$. Therefore, we can get the layer-wise hidden representation $h_s^{1:L}$ via

$$h_s^{1:L} = f_{LM}(\theta_{p_s}, X) \tag{1}$$

For semantic guidance, we directly pass $h_s^{1:L}$ to the alignment unit and denoising unit as new prompt parameters. We use additional two projection layers $W_K$ and $W_V$ to project $h_S^{1:L}$ into semantic hidden states, denoted as $K_s^{1:L}$ and $V_s^{1:L}$.

$$K_s^{1:L} = h_s^{1:L} W_K \\ V_s^{1:L} = h_s^{1:L} W_V \tag{2}$$

## 4.3 Semantic-guided Alignment

This unit generates the target sequence in parallel, by taking a sequence of white noise as input, denoted as $Y_{\text{blank}}$ (a sequence of *<mask>* tokens in our experiments). Similarly, we introduce alignment prompt $\theta_{p_a} = (K_a^{1:L}, V_a^{1:L})$ to efficiently adapt the pre-trained model to generate a target sequence. To grab information from the source sequence, we directly concatenate $(K_a^{1:L}, V_a^{1:L})$ and $(K_s^{1:L}, V_s^{1:L})$, where we get

$$\theta_{p_a} = (\text{concat}([K_s^{1:L}, K_a^{1:L}]), \\ \text{concat}([V_s^{1:L}, V_a^{1:L}])) \tag{3}$$

The alignment output is obtained via:

$$T_0 = f_{LM}(\theta_{p_a}, Y_{\text{blank}}) \tag{4}$$

Formally, we define the alignment loss as $\mathcal{L}_1(\theta_{p_s}, \theta_{p_a})$ given training pair $(X, Y)$ sampling from a dataset. For alignment, we use two variants of non-autoregressive loss to train new parameters, including Connectionist Temporal Classification(CTC) (Libovický and Helcl, 2018) and Mask-Predict (Ghazvininejad et al., 2019). For

constrained generation tasks, specifically, translation, we choose CTC loss objective for its efficiency in speed, as it is a one-step NAR generation method. For free generation tasks, CTC loss performs poorly because free-generation tasks intensify the multi-modality problem (Gu et al., 2018), which we will discuss in Section 5.5. On the contrary, iterative methods choose the best possible modality during early iterations of generation. Therefore, we use Mask-Predict, which is an iterative NAR generation method that sacrifices speed for performance.

## 4.4 Semantic-guided Denoising

Due to the limitation of trainable parameters and the non-autoregressive nature, the generation result of the first-stage alignment is usually far from satisfying. Thanks to the denoising pre-training objective MLM, current language models can be easily adapted to a denoiser. In this step, we also add prompt parameters for denoising $\theta_{p_D} = (K_D^{1:L}, V_D^{1:L})$ to efficiently adapt the understanding model to a language-specific denoiser. Similarly, we get semantic-guided denoising prompt by the following equation:

$$\theta_{p_d} = (\text{concat}([K_s^{1:L}, K_d^{1:L}]), \\ \text{concat}([V_s^{1:L}, V_d^{1:L}])) \tag{5}$$

We take the output sequence in alignment stage as input which is denoted as $T_0$. To avoid overfitting, we add random noise including random deletion or repetition to sequence $\tilde{T}_0 = T_0 + \epsilon$. We can then acquire the denoised logits $T_1$ by:

$$T_1 = f_{LM}(\theta_{p_d}, \tilde{T}_0) \tag{6}$$

We repeat this step and treat $T_1$ as new input to get $T_2$. The loop is running until the output sequence keeps unchanged or we reach the maximum loop number.

For denoising, we use a CTC-based denoiser after the alignment process and adopt the CTC loss $\mathcal{L}_2(\theta_{p_s}, \theta_{p_d})$ given training pair $(Y, T_i)$ where $T_i$ is the output sequence at the i-th step. For translation, the outputs of the alignment stage are directly fed to the denoiser. For other generation tasks, we upsample the alignment result by a factor of 2 by duplicating each token, and then fed the duplicated sequence to the CTC-based denoiser.

## 4.5 Training Objective

The final loss is a combination of the alignment loss and denoising loss by the following equation:

$$\mathcal{L} = \mathcal{L}_1(\theta_{p_s}, \theta_{p_a}) + \mathcal{L}_2(\theta_{p_s}, \theta_{p_d}) \qquad (7)$$

# 5 Experiments

## 5.1 Settings

We run experiments on three multilingual generation datasets including machine translation, question generation, and story generation. Mapping between language codes and full names of all languages used in our paper is presented in Appendix A.

**Dataset**   For experiments on both bilingual and multilingual translation, we use TED dataset (Qi et al., 2018). We focus on English-centric settings and choose 10 languages (Ar, De, Es, Fr, He, It, Ro, Ru, Tr, Vi) with the most training data to construct our multilingual translation task. We choose five additional languages (Kk, Be, Eu, Ms, Bs) with the least training data (less than 6k) for zero-shot cross-lingual evaluation. Details are presented in Appendix B.

For experiments on question generation, we use the Question Generation (QG) split of XGLUE dataset (Liang et al., 2020). Since XGLUE only provides a training set in the English-English direction, we use M2M-100-418M (Fan et al., 2021) to translate the English training set to all other languages. We train all models in the En→X directions and evaluate them on the X→X test sets. For simplicity, we report results on En→En, En→De and En→Fr, where results on En→En represent the monolingual generation ability, and results on En→De, En→Fr represents zero-shot cross-lingual generation ability. For experiments on story generation, we use the Story Generation (SG) split of MTG dataset (Chen et al., 2022). For simplicity, we report monolingual generation result on En→En, and cross-lingual generation results on De→En and Fr→En.

**Implementations**   We use a batch size of 32k to train all transformer models in both AT and NAT. Following (Xu et al., 2021b), we use the transformer-big setting with a learning rate of 5e-4 and a dropout rate of 0.3. We train these models for a maximum of 50 epochs, and average the 5 best checkpoints for inference. We use Fairseq (Ott et al., 2019) for implementation.

For fine-tuning using pre-trained language models like XLM-R and mBART, we use a batch size of 4k tokens and a much smaller learning rate of 3e-5. We train the pre-trained models for a maximum of 80,000 steps. We also use Fairseq.

For adaptation methods on PLMs, we directly follow the hyperparameter setting of (Tan et al., 2022), with a batch size of 32k tokens and a learning rate of 7e-4. We train these models for a maximum of 40,000 steps, and average the 5 best checkpoints for inference. We use THUMT (Tan et al., 2020) for the implementation of the adaptation methods. For translation tasks, the training takes around 40 hours on 8 A100-SXM-80GB GPUs in each translation direction to adapt an XLM-R$_{large}$ model to generators.

**Evaluation Metrics and Hardware**   We calculate case-sensitive **BLEU** (Papineni et al., 2002) using the sacrebleu toolkit (Post, 2018) for translation evaluation [1]. We use **ROUGE-L** (Lin, 2004) for both question generation and story generation. We also report **METEOR** (Banerjee and Lavie, 2005) for story generation. For speed calculation, we average the running time on the test set with batch size set to 1 on a single A100-SXM-80GB GPU, and statistics are averaged by three runs.

**Pre-trained Multilingual Models**   We mainly use three kinds of pre-trained multilingual models in our experiments, including (1) a decoder-only causal model, mGPT (Tan et al., 2022), (2) an encoder-decoder model, mBART (Liu et al., 2020), and (3) an encoder-only model, XLM-R (Conneau et al., 2020).

## 5.2 Baselines

We mainly compare the following baselines in our experiments.

- Transformer (Autoregressive, AT) (Vaswani et al., 2017). We use the transformer-big setting.

- Transformer (Non-autoregressive, NAT) (Libovický and Helcl, 2018). We conduct NAT experiments on Transformer with CTC loss using the transformer-big setting.

- mTransformer. We train a multilingual AT Transformer with 12 encoder layers and 12

---

[1]Signature:nrefs:1|case:mixed|eff:no|tok:13a|smooth:exp|version:2.0.0

| Group | Model | Param. | Speed | Ar→En | De→En | Es→En | Fr→En | He→En | It→En | Ro→En | Ru→En | Tr→En | Vi→En | Avg. |
|---|---|---|---|---|---|---|---|---|---|---|---|---|---|---|
| *Bilingual* | Transformer (AT) | 432M | 1.0× | 32.1 | 36.0 | 41.9 | 40.5 | 38.1 | 38.4 | 35.5 | 24.7 | 26.1 | 27.1 | 34.0 |
| | Transformer (NAT) | 434M | 13.4× | 17.6 | 16.2 | 29.0 | 26.1 | 23.3 | 24.0 | 19.4 | 7.1 | 0.7 | 12.7 | 17.6 |
| *Multilingual* | mTransformer | - | 0.8× | 22.2 | 27.9 | 34.5 | 32.7 | 25.8 | 30.5 | 27.5 | 19.9 | 17.6 | 20.8 | 25.9 |
| | +adapter | 50M | 0.8× | 28.0 | 32.4 | 38.3 | 36.5 | 33.2 | 34.7 | 31.8 | 21.8 | 22.3 | 24.1 | 30.3 |
| *PLM Adaptation* | mGPT + MSP[1] (AT) | 19M | 0.2× | 26.2 | 29.8 | 38.9 | 36.2 | 30.3 | 33.1 | 30.9 | 21.9 | 19.4 | 23.3 | 29.0 |
| | *XLM-R$_{base}$* | | | | | | | | | | | | | |
| | + AT initialization | 390M | 0.9× | 16.6 | 23.5 | 29.5 | 26.7 | 21.0 | 24.9 | 22.7 | 16.1 | 17.6 | 15.7 | 21.4 |
| | + SGA w/o. denoising | 6M | 6.8× | 24.6 | 29.9 | 36.1 | 32.6 | 29.7 | 30.4 | 28.7 | 18.0 | 15.1 | 23.5 | 26.9 |
| | + SGA | 8M | 3.0× | 27.1 | 33.0 | 40.0 | 35.5 | 33.3 | 32.8 | 30.3 | 20.3 | 19.6 | 23.9 | 29.6 |
| | *XLM-R$_{large}$* | | | | | | | | | | | | | |
| | + AT initialization | 960M | 0.6× | 19.2 | 25.7 | 32.4 | 29.9 | 23.4 | 28.4 | 24.9 | 21.5 | 17.4 | 18.3 | 24.1 |
| | + SGA w/o. denoising | 15M | 3.7× | 28.2 | 33.8 | 37.9 | 36.1 | 35.5 | 34.1 | 31.5 | 21.4 | 23.4 | 24.4 | 30.6 |
| | + SGA | 21M | 1.9× | **30.7** | **37.0** | **40.9** | **38.6** | **38.5** | **37.5** | **34.2** | **24.0** | **27.2** | **26.4** | **33.5** |

| Group | Model | Param. | Speed | En→Ar | En→De | En→Es | En→Fr | En→He | En→It | En→Ro | En→Ru | En→Tr | En→Vi | Avg. |
|---|---|---|---|---|---|---|---|---|---|---|---|---|---|---|
| *Bilingual* | Transformer (AT) | 432M | 1.0× | 17.0 | 30.0 | 39.8 | 39.1 | 27.2 | 34.9 | 27.0 | 19.6 | 15.0 | 28.8 | 27.8 |
| | Transformer (NAT) | 434M | 13.4× | 6.2 | 10.6 | 25.2 | 23.0 | 14.6 | 17.5 | 13.1 | 5.3 | 0.4 | 15.2 | 13.1 |
| *Multilingual* | mTransformer | - | 0.8× | 12.3 | 23.6 | 33.1 | 32.2 | 18.9 | 28.4 | 21.7 | 14.8 | 11.1 | 25.2 | 22.1 |
| | +adapter | 50M | 0.8× | 16.3 | 29.3 | 38.9 | 38.4 | 25.6 | 33.6 | 26.3 | 19.1 | 15.2 | 30.3 | 27.3 |
| *PLM Adaptation* | mGPT + MSP (AT) | 19M | 0.2× | 11.6 | 24.1 | 31.7 | 32.3 | 20.7 | **29.6** | 19.2 | **17.9** | 11.7 | 24.4 | 22.3 |
| | *XLM-R$_{base}$* | | | | | | | | | | | | | |
| | + AT initialization | 390M | 0.9× | 7.9 | 17.6 | 26.4 | 22.6 | 14.1 | 20.1 | 15.8 | 10.6 | 6.9 | 18.4 | 16.0 |
| | + SGA w/o. denoising | 6M | 6.8× | 8.7 | 19.4 | 28.8 | 23.8 | 16.9 | 25.6 | 18.8 | 11.5 | 7.2 | 22.8 | 18.4 |
| | + SGA | 8M | 3.0× | 11.1 | 21.7 | 32.3 | 27.5 | 18.9 | 28.5 | 21.5 | 14.7 | 8.4 | 24.3 | 20.9 |
| | *XLM-R$_{large}$* | | | | | | | | | | | | | |
| | + AT initialization | 960M | 0.6× | 9.9 | 19.8 | 29.3 | 26.2 | 17.4 | 23.1 | 18.0 | 12.2 | 11.5 | 26.0 | 19.3 |
| | + SGA w/o. denoising | 15M | 3.7× | 11.3 | 22.2 | 33.4 | 30.6 | 19.5 | 26.5 | 22.0 | 13.1 | 9.9 | 24.7 | 21.3 |
| | + SGA | 21M | 1.9× | **13.1** | **25.2** | **37.1** | **34.3** | **21.3** | 29.2 | **24.6** | 15.5 | **11.8** | **26.9** | **23.9** |

Table 1: Results of X→EN and EN→X translation. "Param." represents the total number of trainable parameters. "Speed" represents the inference speed when batch size is 1. Scores in bold represent the best performance in the *Adapt PLM* setting. Compared with the traditional fine-tuning method that directly adopts XLM-R as initialization, SGA brings large performance gains, with 8.2 BLUE on XLM-R$_{base}$ and 9.4 BLUE on XLM-R$_{large}$ on X→EN, and with 4.9 BLEU on XLM-R$_{base}$ and 4.6 BLEU on XLM-R$_{large}$ on EN→X, showing the effectiveness of SGA on adapting multilingual understanding models to multilingual generators.

decoder layers on the TED multilingual translation datasets. Other hyperparameters are shared with Transformer-big. To report X→En and En→X results, we train two mTransformer models using all X→En and En→X data in TED, respectively.

- mTransformer + adapter. We use language-specific adapters (Bapna and Firat, 2019) on top of our trained mTransformer. We append adapters to both encoder layers and decoder layers and use a feed-forward layer dim of 1,024, which finally results in 50M extra parameters for each language pair.

- mGPT + MSP (Tan et al., 2022). mGPT + MSP introduces multi-stage prompting over a multilingual GPT model with 560M parameters. We implement this baseline following the same setting as the original paper.

- XLM-R w. AT initialization. Under this setting, we initialize the encoder of an autoregressive Transformer with the weights of XLM-R, and fine-tune the whole parameters. We use two variants of XLM-R: XLM-R$_{base}$ with 270M parameters, and XLM-R$_{large}$ with 550M parameters.

## 5.3 Main Results

**SGA achieves large performance improvements over traditional initialization-based adaptation** Table 1 presents the multilingual translation experiments. We find that initializing an autoregressive Transformer model from XLM-R only brings slight improvements by comparing with Transformer (NAT). We speculate the different nature of AR and NAR leads to performance degradation when using XLM-R as initializers. Compared with the traditional fine-tuning method that directly adopts XLM-R as initialization, SGA brings large performance gains, with 8.2 BLUE on XLM-R$_{base}$ and 9.4 BLUE on XLM-R$_{large}$ on X→EN, and with 4.9 BLEU on XLM-R$_{base}$ and 4.6 BLEU on XLM-R$_{large}$ on EN→X, showing the effectiveness of SGA on adapting multilingual understanding models to multilingual generators. With 21M trainable parameters, our method achieves comparable performance with bilingual counterparts and even better performance in several language directions (De, He, Tr). The bottom part presents translation results on the **En→X** directions.

**SGA shows better efficient inference over adaptation baselines** Compared with the original mTransformer baseline including the adapter setting, our method achieves $1.9/0.8 = 2.4\times$ speedups with better performance. As a compari-

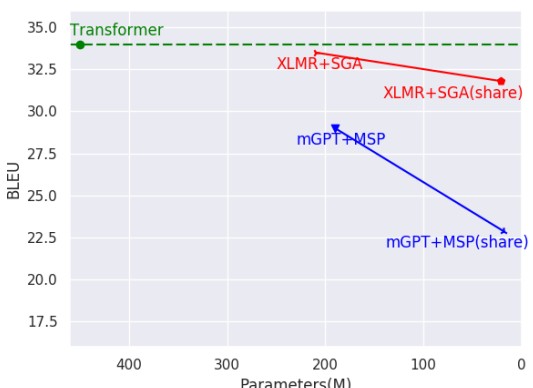

Figure 3: Tradeoff between parameter size and BLEU scores. "share" represents prompt sharing. Our proposed XLM-R+SGA with prompt sharing strategy can further reduce parameters without sacrificing much of the performance. As a comparison, mGPT + MSP drops significantly.

son, mGPT + MSP brings higher inference latency due to multi-stage prompting and AT decoding.

**Denoising brings large performance gains in all directions** On both XLM-R$_{base}$ and XLM-R$_{large}$, our proposed denoising technique brings an average gain of 2.8 BLEU by increasing a very small amount of parameters. This confirms our conjecture that multilingual understanding models can be parameter-efficient language-specific denoisers due to the denoising pretraining nature of MLM.

**XLM-R boots the performance of NAT** Existing NAT model (NAT+CTC) produces poor results in all language directions. It is because NAT generally requires an AT model to generate distillation datasets (Gu et al., 2018) which we do not provide in this paper. The NAR nature of XLM-R makes it possible to boost NAT performance. With SGA, XLM-R outperforms NAT baselines by a large margin, indicating that NAR pre-training can be further explored in future work to make multilingual understanding models better generators.

### 5.4 Prompt Sharing Analysis

We further reveal the potential of our proposed SGA by sharing prompts across languages. We combine datasets in the 10 X→En language directions selected in Section 5.1, and compare perfor-

---

[1]MSP uses different tokenization processing scripts for evaluation. To have a fair comparison, we reproduce the mGPT + MSP results in all language directions based on their public code.

mance with multilingual Transformer and mGPT + MSP, mBART. All baselines including mTransformer are trained using the combined dataset.

**Prompt sharing enables a compact X→En translation plugger** SGA achieves a better tradeoff between parameter size and inference performance by sharing prompts across all X→En directions, which achieves competitive performance with the bilingual Transformer. As adaptation methods, Figure 3 presents the tradeoff between parameter size and performance. Performance is evaluated by averaging the test set BLEU in all directions. SGA achieves a better tradeoff performance than the AT counterpart, mGPT + MSP. With only 21M parameters, SGA enables a multilingual understanding LM a unified and impressive X→En translator.

**Prompt sharing brings impressive zero-shot cross-lingual transfer** Sharing prompts also empower SGA with strong zero-shot cross-lingual transfer ability. We choose 5 languages (Kk, Be, Eu, Ms, and Bs) with the least training data in X→En directions in the TED dataset, and compare performance with multilingual Transformer, mGPT + MSP and mBART. Table 2 presents the zero-shot cross-lingual transfer ability. (i) Trained only in 10 language directions, mTransformer outperforms mGPT + MSP in the supervised language directions with a large performance gap, while our method, XLM-R$_{large}$+SGA is still superior to both methods. (2) XLM-R+SGA achieves good performance in zero-shot X→En experiments, with a substantial performance improvement when compared with mGPT + MSP and mTransformer. (3) Although still lags behind the performance of mBART on supervised language directions, mBART supports much fewer languages than XLM-R (50 vs. 100), which presents limitations in zero-shot cross-lingual performance.

### 5.5 Other Generation Scenarios

In this section, we test the performance of our model in various generation tasks other than multilingual translation to further explore the generation ability of multilingual understanding models. Table 3 presents the results of both monolingual and cross-lingual results on question generation and story generation. For both tasks, we provide a monolingual result in the En→En direction for reference. For the question generation task on XGLUE, since it only provides test sets in the

| Group | Model | Param. | Supervised | | | | Unsupervised | | | | |
|---|---|---|---|---|---|---|---|---|---|---|---|
| | | | De | Es | Fr | It | Kk | Be* | Eu* | Ms* | Bs* |
| *Multilingual* | mTransformer | 432M | 32.5 | 39.2 | 37.5 | 35.5 | 1.2 | 2.1 | 2.1 | 1.4 | 1.8 |
| | mBART-fine-tune | 610M | **41.0** | **46.3** | **44.4** | **42.8** | **13.6** | 2.2 | 1.6 | 21.0 | 16.5 |
| *Adapt PLM* | mGPT + MSP | 19M | 27.6 | 35.2 | 33.2 | 32.0 | 6.7 | 17.1 | 10.7 | 19.3 | 14.3 |
| | $XLM\text{-}R_{base}$ + SGA | 8M | 31.9 | 37.5 | 36.0 | 33.9 | 7.2 | 19.6 | 12.9 | 24.0 | 27.0 |
| | $XLM\text{-}R_{large}$ + SGA | 21M | 34.2 | 39.3 | 38.3 | 36.2 | 11.4 | **26.8** | **20.7** | **30.1** | **34.0** |

Table 2: Zero-shot translation performance on TED in the X→En directions. Our method achieves impressive performance in the zero-shot cross-lingual setting, with significant improvement in all unsupervised translation directions compared to mGPT + MSP. * represents that this language is not supported in mBART.

| Models | Para. | Speed | Question Generation | | | Story Generation | | | | | |
|---|---|---|---|---|---|---|---|---|---|---|---|
| | | | En→En | En→De* | En→Fr* | En→En | | De→En | | Fr→En | |
| | | | RL↑ | RL↑ | RL↑ | RL↑ | Meteor↑ | RL↑ | Meteor↑ | RL↑ | Meteor↑ |
| mGPT + MSP | 19M | 1.0× | **36.3** | **17.5** | 17.7 | **16.7** | **16.1** | 13.7 | 13.7 | 14.2 | 13.8 |
| $XLM\text{-}R_{large}$ | | | | | | | | | | | |
| + AT initialization | 960M | 1.7× | 28.9 | 9.1 | 10.5 | 8.4 | 9.1 | 9.6 | 9.5 | 9.3 | 10.3 |
| + SGA w/o. denoising | 15M | 4.5× | 34.4 | 16.9 | 19.2 | 15.7 | 15.5 | 14.3 | 14.4 | **15.3** | 13.9 |
| + SGA | 21M | 4.3× | 35.6 | 17.4 | **19.9** | 15.5 | 15.9 | **14.4** | **15.0** | 14.8 | **14.6** |

Table 3: Results on question generation and story generation. RL represents the F1-score of Rouge-L. * represents zero-shot cross-lingual scenarios. SGA beats initialization-based methods on XLM-R in all cross-lingual scenarios with a substantial improvement, and achieves comparable results with mGPT + MSP.

X→X directions, we train all models on the training set of En→X directions, and evaluate the model performance on the X→X directions for zero-shot cross-lingual generation. For the story generation task on MTG, we test supervised cross-lingual generation performance on the X→En direction. We report Rouge-L scores for both tasks, and report METEOR scores additionally for story generation.

Table 3 presents the generation results. For free generation tasks, we use Mask-Predict for the alignment stage, and we set the iteration number to 4 in this table. (i) Our proposed method XLM-R+SGA can achieve comparable performance while notable acceleration, when compared with an autoregressive-based model, mGPT + MSP, on almost all generation tasks. (ii) Using XLM-R to initialize an autoregressive Transformer totally loses the zero-shot cross-lingual ability. Although it performs moderately on the supervised monolingual direction (En→En) on Question Generation, it performs poorly on the zero-shot directions including En→De and En→Fr. (iii) Our denoising technique is proven helpful in further improving the generation quality in both tasks without sacrificing much of the speed.

**Tradeoff between iterative prediction and CTC-based denoising in free generation tasks** For free-generation tasks, unlike translation, we use iterative mask prediction instead of CTC for the

| Group | # Iter. | Meteor↑ | Speed |
|---|---|---|---|
| *w/o. denoising* | 2 | 14.5 | 14.1 sent/s |
| | 4 | 15.5 | 8.3 sent/s |
| | 8 | 15.8 | 5.2 sent/s |
| *w. denoising* | 0 | 13.0 | 21.6 sent/s |
| | 2 | 15.3 | 11.8 sent/s |
| | 4 | 15.9 | 7.9 sent/s |
| | 8 | 16.1 | 4.8 sent/s |

Table 4: Trade-off between CTC-based denoiser and number of iterations on En→En generation on story generation. Batch size is set to 1. Denoising brings better performance and presents a better tradeoff between performance and inference speed.

alignment stage. Free generation introduces much more modalities than constrained generation tasks, specifically, translation, which intensifies the multi-modality problem in NAR generation (Gu et al., 2018). Therefore, we use an iterative method, Mask-Predict, to improve the generation quality for the alignment stage of our proposed SGA.

Although increasing the iteration number in the alignment stage can obviously lead to better performance, it will also intensify the latency problem. Our CTC-based denoiser can not only bring better performance, but also a better tradeoff between performance and speed, which is presented in Table 4. When the iterations of the alignment stage is set to the same, using the CTC-based denoiser leads to

better performance with a slight sacrifice in speed. Using CTC with 4-step decoding can outperform 8-step decoding both in performance and speed. However, using CTC alignment alone will lead to inferior performance (0-step decoding) because of the multi-modality problem.

# 6 Conclusion

In this paper, we propose an effective approach to adapt existing pre-trained multilingual understanding models to multilingual generators. On translation tasks, experiments demonstrated that our proposed method achieves large performance improvements and notable acceleration with strong cross-lingual generation ability. On free-generation tasks including question generation and story generation, our method also achieves comparable performance with AT-based method with impressive speedups. Although still lagging behind pretrained multilingual AT models (e.g., mBART) in supervised fine-tuning settings in translation, our proposed method show better zero-shot abilities and faster inference.

# 7 Limitations

Although our proposed method has achieved notable speedups and performance improvements in the multilingual setting, we still lag behind in bilingual translation, especially in high-resource scenarios. In addition, there still remains a gap between NAR pre-trained models and AR pre-trained models. Generally, XLM-R with fine-tuning is largely inferior to mBART fine-tuning. Despite the gap can be largely reduced with our method, the gap still exists. In future work, we would like to explore pre-training methods to make pretrained multilingual NAR models better generators.

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

## A  Language Code References

We provide the list of languages and corresponding language codes used in our experiments in Table 5.

| Name | Arabic | German | Spanish | French |
|---|---|---|---|---|
| Code | Ar | De | Es | Fr |

| Name | Hebrew | Italian | Romanian | Russian |
|---|---|---|---|---|
| Code | He | It | Ro | Ru |

| Name | Turkish | Vietnamese | Kazakh | Belarusian |
|---|---|---|---|---|
| Code | Tr | Vi | Kk | Be |

| Name | Basque | Malay | Bosnian | - |
|---|---|---|---|---|
| Code | Eu | Ms | Bs | - |

Table 5: Full names and corresponding codes of languages used in our experiments.

## B  Details of TED Dataset

Table 6 presents a rough statistics number of the chosen 15 languages in our main experiment.

| Name | Ar | De | Es | Fr | He |
|---|---|---|---|---|---|
| Num. | 211k | 165k | 193k | 189k | 208k |

| Name | It | Ro | Ru | Tr | Vi |
|---|---|---|---|---|---|
| Num. | 201k | 178k | 205k | 180k | 169k |

| Name | Kk | Be | Eu | Ms | Bs |
|---|---|---|---|---|---|
| Num. | 3,234 | 4,392 | 5,094 | 5,104 | 5,566 |

Table 6: A rough statistics of the chosen 15 languages (10 for supervised setting and 5 for zero-shot cross-lingual setting) for the number of train samples in TED dataset.