# OpenReview forum: "Extrapolating Multilingual Understanding Models as Multilingual Generators"
_EMNLP/2023/Conference — EMNLP 2023 Findings_

### Official Review · Reviewer_Ne93 · 2023-08-01

**Soundness:** 4

**Excitement:**

4: Strong: This paper deepens the understanding of some phenomenon or lowers the barriers to an existing research direction.

**Paper Topic And Main Contributions:**

This paper explores methods to empower multilingual understanding models (encoder) the generation abilities to get a unified model.

Specifically, they started from a multilingual encoder (XLM-R) and propose a Semantic-Guided Alignment-then-Denoising (SGA) approach to adapt an encoder to a multilingual generator with a small number of new parameters. Experimental results on various tasks including machine translation, question generation demonstrate the effectiveness of their proposed method.

Except the above results, they also observed that an interesting phenomenon: XLM-R is still inferior to mBART in supervised settings despite better results on zero-shot settings.

**Questions For The Authors:**

Questions:

(1) I'm not an expert in machine translation, do you know why the BLEU score pf X->en is much better than en->X for most languages (in Table 1)? any intuitive explanation for this?

(2) in Table 1, row 2, Transformer (NAT), which model do you refer to? I think there're so many NAT models. Is this NAT model STOA?


**Reasons To Accept:**

This paper tackle an interesting topic and present a good solution to it, the experimental results look like strong.

**Reasons To Reject:**

I didn't find any obvious reasons to reject this paper directly.

**Reproducibility:**

3: Could reproduce the results with some difficulty. The settings of parameters are underspecified or subjectively determined; the training/evaluation data are not widely available.

**Reviewer Confidence:**

2: Willing to defend my evaluation, but it is fairly likely that I missed some details, didn't understand some central points, or can't be sure about the novelty of the work.

---

> ### Author Rebuttal · Authors · 2023-08-28
>
> Thanks for your review and we appreciate your approval of our work.
>
> >**Question 1: Why the BLEU score pf X->en is much better than en->X for most languages (in Table 1)?**
>
> The generation ability of XLM-R is consistent with the understanding ability of XLM-R.
>
> On understanding tasks, XLM-R also presents strong ability in English, Spanish, and French,  while inferior performance in other languages. Therefore, adapting XLM-R to English/Spanish/French decoders often yields better performance than adapting XLM-R to other language decoders.
>
> >**Question 2: Is the NAT model in Table1, row 2 STOA?**
>
> The NAT model presented in Table 1 is introduced in [1]. It is not the SOTA model but a classical one.
>
> In this paper, we intend to provide an efficient PLM adaptation method for pre-trained multilingual encoders, while previous NAT works focus on bilingual settings that are trained from scratch.
>
> Therefore we do not include other NAT models but only a representative baseline in Table 1.
>
>
> [1] Non-Autoregressive Neural Machine Translation (Gu et al., ICLR 2018)

---

### Official Review · Reviewer_qScF · 2023-08-02

**Typos Grammar Style And Presentation Improvements:** Line 271
**Soundness:** 4

**Excitement:**

4: Strong: This paper deepens the understanding of some phenomenon or lowers the barriers to an existing research direction.

**Missing References:**

N/A

**Paper Topic And Main Contributions:**

There are significant advantages to non-autoregressive generation in terms of both generation efficiency and flexibility. This paper is focused on proposing an efficient adaptation method to improve the generation quality of multilingual understanding models in a parameter-efficient way. The paper also presents a diffusion-like approach -- semantic-guided denoiser to further enhance the generation quality. Extensive experiments demonstrate that their proposed method outperforms the traditional methods by a large margin.

**Questions For The Authors:**

1) Why are the results of mTransformer on the same language direction different between Table 1 and Table 2? Why are the parameters of mTransformer not labeled in Table 1?
2) The author seems to have not introduced how to deal with the problem of inconsistent input and output lengths, such as how to generate a target sentence shorter or longer than the source sentence?

**Reasons To Accept:**

1. The authors propose a novel approach to transform non-autoregressive generation into a step-by-step denoising process, utilizing the denoising ability of a multilingual pre-trained understanding model for implementation.

2. Authors adopt prompt-based approaches to achieve high parameter-efficiency during semantic alignment and denoising procedures.

3. The work is solid. Experiments demonstrate that the proposed method can significantly improve the performance of non-autoregressive generation with only a small number of parameters required. Additionally, this method can be extended to many other generation tasks and has better zero-shot knowledge transfer capability.

**Reasons To Reject:**

1. Some parts are not very clear, which caused me confusion:
  a) Why are the results of mTransformer on the same language direction different between Table 1 and Table 2? Why are the parameters of mTransformer not labeled in Table 1?
  b) The author seems to have not introduced how to deal with the problem of inconsistent input and output lengths, such as how to generate a target sentence shorter or longer than the source sentence?

2. It is not clear whether authors will release their codes and models.

**Reproducibility:**

3: Could reproduce the results with some difficulty. The settings of parameters are underspecified or subjectively determined; the training/evaluation data are not widely available.

**Reviewer Confidence:**

4: Quite sure. I tried to check the important points carefully. It's unlikely, though conceivable, that I missed something that should affect my ratings.

---

> ### Author Rebuttal · Authors · 2023-08-28
>
> Thanks for your review, and we appreciate your approval of our work.
>
> >**Question 1: Why are the results of mTransformer on the same language direction different between Table 1 and Table 2? Why are the parameters of mTransformer not labeled in Table 1?**
>
> Because of the training setting, which are different in Table 1 and Table 2.
>
> In Table 1, we use different soft prompts for each language direction, which represents bi-lingual translation results.
>
> In Table 2, we focus more on the zero-shot cross-lingual ability, therefore we use a multilingual version of the soft prompt, i.e. the soft prompt is shared across all X $\rightarrow$ En directions.
>
> We do not label the parameters of mTransformer in Table 1, only label the number of tuned parameters in each baseline.
>
> Because our method belongs to the adaption setting. mTransformer and XLM-R are pre-trained models and none of mTransformer parameters are fine-tuned in each language direction.
>
> >**Question 2: How to generate a target sentence shorter or longer than the source sentence?**
>
> For generating a target sentence with inconsistent length with the source sentence, in our experiments, we follow the classical approach introduced in CTC[1]. In detail, we up-sample the length of the source sentence by a factor of two and use CTC loss to convert the optimization of token distributions to path distributions. Thanks for pointing out the writing issue. We will add a more detailed description in our revision.
>
> >**Weakness 2: It is not clear whether the authors will release their codes and models.**
>
> We will release the code and the models.
>
> [1] End-to-End Non-Autoregressive Neural Machine Translation with Connectionist Temporal Classification (Jindřich Libovický, Jindřich Helcl, EMNLP2018)

---

### Official Review · Reviewer_5rFu · 2023-08-02

**Typos Grammar Style And Presentation Improvements:** 73
**Soundness:** 3

**Excitement:**

2: Mediocre: This paper makes marginal contributions (vs non-contemporaneous work), so I would rather not see it in the conference.

**Missing References:**

See "the reasons to reject" section.

**Paper Topic And Main Contributions:**

This paper proposes a three-stage approach to adapt multilingual encoders for generation tasks. During the first stage, the authors train soft prompts to allow a pre-trained encoder to extract representations; During the second stage, the authors train another series of soft prompts for the same encoder model to do alignment to the target sequence; During the third stage, train another series of soft prompts that iteratively refines the aligned sequence to the target sequence. Experiments on Machine Translation, Question Generation and Story Generation show that their method outperforms training decoder-only models from scratch and and naive soft prompt tuning on multilingual GPT models.

**Questions For The Authors:**

First see the "reasons to reject" section.

A. Line 76. What does "complicated" mean right here? If there is a comparison of difficulty, please mention machine translation is complicated than what task.

B. What is AT and NAT abbreviation for? Should it be AR and NAR?


**Reasons To Accept:**

1. The problem that this paper studies is important: how to modify encoder only models for generations as the non-autoregressive pre-train objective is hard to adapt for generation. The method is simple and generalizable to various tasks since we only need to train a few set of soft prompts.

2. The experiments show that their method can outperform simply initializing the encoder with a pre-trained model, direct training a encoder-decoder model on the task albeit training on much less parameters due to the parameter efficient fine-tuning.

3. The generation speed is accelerated because compared to decoder only models which needs to pass the generated output many times through the model, encoder only model generates multiple tokens simultaneously and iteratively denoises them.

**Reasons To Reject:**

1. This paper misses some important work in their discussion: The authors fail to compare with or at least cite some important work that utilizes pre-trained encoders for generation. e.g. BiBERT[1], which is the current state-of-the art on IWSLT 14 De-En translation. The authors also compare to mBART and show that although their method underperforms, they win in generation speed and trained parameters. I want to say that they should compare to doing prompt tuning on mBART as well to make it a fair comparison between number of trained parameters. If it outperforms, then it strengthens their claim.

2. Motivation Unclear: The authors mention that an important reason to use encoders for generation is because of their speed. However, works [2] have demonstrated that standard encoder-decoder models can also be fast and comparable in performance by using a large encoder and a small decoder. On the other hand, if the argument is that encoder models excel in NLU tasks, then using Pattern Exploit Training ,encoder-decoder models can be comparable to standard encoder models in NLU tasks [3]. My point is, I want to see a more motivating argument for studying how to adapt encoder models for generation aside from performance and efficiency.

3. The paper is not very well written. It does not have a clear distinction between model architecture (encoder, decoder, enc-dec) and training objective (autoregressive, auto-encoding). They use autoregressive (AR) and encoders interchangeably - which should be mentioned in the paper. Some of the facts need modifying: e.g. the authors mention that unlike AR models which can only generate from left to right, NAR models as the advantage of generating in arbitrary order. I don't think this is true since there has been work that generates with permuted orders with encoder-decoder models [4].


[1] BERT, mBERT, or BiBERT? A Study on Contextualized Embeddings for Neural Machine Translation (Xu et al., EMNLP 2021)

[2] Deep Encoder, Shallow Decoder: Reevaluating Non-autoregressive Machine Translation (Kasai et al., ICLR 2021)

[3] Exploring the Limits of Transfer Learning with a Unified Text-to-Text Transformer (Raffel et al., JMLR 2020)

[4] GLM: General Language Model Pretraining with Autoregressive Blank Infilling (Du et al., ACL 2022)


**Reproducibility:**

4: Could mostly reproduce the results, but there may be some variation because of sample variance or minor variations in their interpretation of the protocol or method.

**Reviewer Confidence:**

4: Quite sure. I tried to check the important points carefully. It's unlikely, though conceivable, that I missed something that should affect my ratings.

---

> ### Author Rebuttal · Authors · 2023-08-28
>
> Thanks for your review and valuable suggestions. Thanks for pointing out the typos and writing issues, and we will fix them accordingly.
>
> >**Weakness 1-1: This paper misses some important work in their discussion, e.g., BiBERT**
>
> Thanks for pointing out BiBERT.  Although BiBERT also utilizes pre-trained encoders for generation, it is quite different from our works from the following perspectives.
>
> (a) BiBERT utilizes pre-trained encoders by initializing Transformer models with parameters of a bilingual BERT, while our work focuses on effective adaptation methods on pre-trained encoders.
>
> (b) BiBERT is still an autoregressive translation model, while we explore the non-autoregressive nature of pre-trained encoders.
>
> (c) BiBERT focuses on a bilingual setting, while our method targets on multilingual setting.
>
> Therefore, we do not include BiBERT in our experimental baselines in this version. We are glad to add this discussion about BiBERT in our next revision.
>
> >**Weakness 1-2: The authors should compare to doing prompt tuning on mBART as well to make it a fair comparison between the number of trained parameters.**
>
> For a more comprehensive comparison with mBART, we agree that incorporating a baseline that employs prompt tuning on mBART would be beneficial, and we intend to include it in the next revision.
>
> In this revision, the comparison between mBART with our model focuses solely on speed, which is a fair comparison.  The speed improvement is consistent since prompt tuning does not change the auto-regressive decoding nature of mBART.
>
> >**Weakness 2-1: Motivation Unclear. Works have demonstrated that standard encoder-decoder models can also be fast and comparable in performance by using a large encoder and a small decoder.**
>
> Although shallow decoder models can accelerate inference by reducing the decoding layers, the acceleration is orthogonal to the decoding strategy (AR or NAR). Works have pointed out that NAR methods could also benefit from decreasing the decoding layers [1].
>
> >**Weakness 2-2: The reviewer wants to see a more motivating argument for studying how to adapt encoder models for generation aside from performance and efficiency.**
>
> Apart from performance and efficiency, our other motivation is the adaptation of foundation models.
>
> How to adapt existing foundation models for general usage including translation is an important research direction recently.
>
> Many works have been devoted to adapting generative models to understanding tasks, but how to adapt an understanding model to generative tasks remains unexplored. Given that XLM-R has already been scaled up to 11B, an xx-large version, we would like to explore the possibility of exploring the general usage of understanding models apart from understanding tasks.
>
> >**Weakness 3-1: The paper is not very well written. It does not have a clear distinction between model architecture (encoder, decoder, enc-dec) and training objective (autoregressive, auto-encoding).**
>
> Thanks for pointing out the writing issues, and we will fix them in the revision.
>
> >**Weakness 3-2: Encoder-decoder models can generate with permuted orders.**
>
> The generation ability of decoder models or encoder-decoder models are limited in infilling tasks with strict length requirement, e.g. filling blanks with a restricted number of words or writing acrostics.
>
> Encoder-based models can naturally cope with tasks by pre-defining a given number of mask tokens, but autoregressive models (decoder only and enc-dec) are known to be weak in controlling sequence lengths without massive instruction tuning.
>
> >**Question A: Line 76. What does "complicated" mean right here? If there is a comparison of difficulty, please mention machine translation is more complicated than what task.**
>
> By "complicated", we meant to clarify that existing learning-free methods perform inferior in generation tasks that require strict semantic alignment as well as word alignment and high generation quality, i.e. machine translation.
>
> Current learning-free or energy-based models, e.g. Mix and Match[2], are only proved effective on free generation tasks that only require semantic alignment, e.g. controllable debiasing. We will change "complicated" to a more appropriate word in future revisions.
>
> >**Question B: What is AT and NAT abbreviation for? Should it be AR and NAR?**
>
> In our paper, we use AT to represent autoregressive translation, while NAT is for non-autoregressive translation. AR and NAR are decoding strategies.
>
> [1] Non-Autoregressive Neural Machine Translation: A Call for Clarity (Schmidt et al., EMNLP2022)
>
> [2] Mix and Match: Learning-free Controllable Text Generationusing Energy Language Models(Mireshghallah et al., ACL2022)

---

### Meta-Review · Area_Chair_PzXe · 2023-09-19

**Recommendation:** 4

**Metareview:**

This paper explore to transform a encoder only model (in this case an XLM-R) into a generator model. This aims to use these models multilingual encoding capabilities. It is done with a “Semantic-Guided Alignment-then-Denoising (SGA) approach to adapt an encoder to a multilingual generator”. This allows the model to use a small amount of new parameters for this adaption. The reviewers agree that the tackled problems is relevant to the field, and that the proposal is exciting. The main issues found where related with a unclear writing and some terminology confusions. Also the related work should be improved.

---

### Decision · Program_Chairs · 2023-10-07

**Decision:**

Accept-Findings

**Comment:**

This paper explore to transform a encoder only model (in this case an XLM-R) into a generator model. This aims to use these models multilingual encoding capabilities. It is done with a “Semantic-Guided Alignment-then-Denoising (SGA) approach to adapt an encoder to a multilingual generator”. This allows the model to use a small amount of new parameters for this adaption. The reviewers agree that the tackled problems is relevant to the field, and that the proposal is exciting. The main issues found where related with a unclear writing and some terminology confusions. Also the related work should be improved.